# Towards Understanding Multimodal Fine-Tuning: A Case Study into Spatial Features

## Abstract

Contemporary Vision–Language Models (VLMs) achieve strong performance on a wide range of tasks by pairing a vision encoder with a pre-trained language model, fine-tuned for visual–text inputs. Yet despite these gains, it remains unclear how language backbone representations adapt during multimodal training and when vision-specific capabilities emerge. In this work, we present the first mechanistic analysis of VLMs adaptation process. Using stage-wise model diffing, a technique that isolates representational changes introduced during multimodal fine-tuning, we reveal how a language model learns to "see". We first identify vision-preferring features that emerge or reorient during fine-tuning. We then show that a selective subset of these features reliably encodes spatial relations, revealed through controlled shifts to spatial prompts. Finally, we trace the causal activation of these features to a small group of attention heads. Our findings show that stage-wise model diffing reveals when and where spatially-grounded multimodal features arise. It also provides a clearer view of modality fusion by showing how visual grounding reshapes features that were previously text-only. This methodology enhances the interpretability of multimodal training and provides a foundation for understanding and refining how pretrained language models acquire vision-grounded capabilities.

## 1 Introduction

Large vision–language models (VLMs) have achieved strong performance on multimodal tasks, including visual question answering (VQA), image captioning, object detection, and visual grounding (Li et al., 2024; AI, 2024). These gains are typically realized by fine-tuning pretrained language models to process visual inputs through projected token sequences, allowing for seamless fusion of image and text representations (Xu et al., 2024; Zhang et al., 2024; Dong et al., 2024b;a). Yet we lack a mechanistic account of how language representations adapt during multimodal training and when vision-specific capabilities emerge (Khayatan et al., 2025; Venhoff et al., 2025c; Stan et al., 2024).

In this work, we introduce a method for analyzing multimodal adaptation in VLMs through stage-wise model diffing (Bricken et al., 2024). This mechanistic interpretability technique isolates representational changes introduced during fine-tuning by comparing sparse autoencoder (SAE) dictionaries across training stages, models, or datasets. By tracking how features rotate, emerge, or are repurposed, it has been shown to uncover subtle shifts such as sleeper-agent features (Hubinger et al., 2024). We extend this approach to the multimodal setting, presenting the first application of stage-wise model diffing to study how pretrained language features evolve under visual grounding.

Concretely, we fine-tune LLaMA-Scope SAEs on activations extracted from the LLaVA-More model (He et al., 2024) on 50k VQAv2 dataset samples(Goyal et al., 2017). This warm-start preserves the original feature basis while adapting to multimodal activations. We isolate features that gain visual preference and undergo strong geometric rotation, serving as anchors for studying spatial representations in the backbone. To identify which adapted features encode spatial reasoning, we apply a controlled dataset shift from general VQA to spatial queries. Features that are preferentially recruited under spatial prompts form a selective subset, which we validate through automatic and manual interpretation. These features consistently activate on questions about object placement, relative position, and orientation. Figure 2 highlights the filtered spatial features.

Finally, we use attribution patching to trace the causal pathways by which these spatial features are activated. Our results reveal a sparse set of mid-layer heads that consistently drive spatial

representations, often localizing to semantically meaningful regions and reappearing across related prompts. These findings support the hypothesis that a small number of specialized attention heads coordinate visual grounding within the model. Our contributions are as follows:

- We extend stage-wise model diffing to the multimodal setting, providing the first feature-level account of how pretrained language backbones adapt under visual grounding.
- We introduce a systematic pipeline to isolate adapted features, identify those selectively recruited by spatial queries, and filter out lexical artifacts.
- We show that these spatially selective SAE features are functionally involved in reasoning, through empirical evidence and ablation studies, supported by interpretive checks.
- We causally attribute the emergence of spatial features to a small subset of attention heads using scalable attribution patching, highlighting structured pathways for visual grounding.

By focusing on feature-level change, our approach complements high-level alignment analyses and probing-based methods, providing a deeper mechanistic view of how models "learn to see". More broadly, this work offers a framework for auditing and refining multimodal training regimes, with implications for safety-critical domains and targeted fine-tuning in specialized applications.

## 2 RELATED WORK

**Model Diffing and Representation Dynamics** Model diffing techniques aim to isolate how internal representations change across models or training stages. Early work focused on coarse similarity measures, such as visualizing function-space geometry (Olah, 2015; Erhan et al., 2010), stitching intermediate layers across models (Lenc & Vedaldi, 2015; Bansal et al., 2021), or defining new similarity metrics (Kornblith et al., 2019; Barannikov et al., 2021). Later studies examined alignment at the level of individual neurons, showing convergent units across independently trained networks (Li et al., 2015; Olah et al., 2020).

Sparse autoencoders (SAEs) offered a feature-level lens, and prior work (Kissane et al., 2024) showed that SAEs largely transfer between base and fine-tuned models, implying most features are preserved and only a minority are altered. This motivates methods that can isolate and precisely interpret those changes. Stage-wise model diffing (Bricken et al., 2024) offers such fine-grained resolution, revealing sleeper-agent features and distinguishing between base and chat-tuned models (Minder et al., 2025a). Extensions to multimodal models highlight similar representational shifts, with concept-shift vectors proposed for steering (Khayatan et al., 2025) and evidence that alignment converges in middle-to-late layers (Venhoff et al., 2025c). These remain semantic-level analyses, whereas our work applies stage-wise diffing with SAEs to the backbone, giving the first mechanistic account of multimodal fine-tuning, showing how it rotates features and induces spatial grounding in pretrained language models.

**Multimodal Mechanistic Interpretability.** Compared to the rapidly growing literature on mechanistic interpretability of textual LLMs, relatively few studies have examined the internal mechanisms of multimodal large language models (MLLMs). Existing work falls into two main categories.

First, tool-based and causal analyses aim to explain model behavior at a high level. Approaches include interpretability toolkits based on attention patterns, relevancy maps, and causal interventions (Stan et al., 2024). Other work uses interventions to trace how information is stored and transferred (Basu et al., 2024), or applies causal mediation to study how BLIP integrates visual evidence (Palit et al., 2023). Second, probing-based studies focus on the representations themselves. Several works analyzed CLIP, identifying both strengths and limitations (Tong et al., 2024; Gandelsman et al., 2023; Chen et al., 2023). Others reported multimodal neurons responsive to joint visual–textual concepts (Schwettmann et al., 2023) and examined how VLMs differentiate hallucinated from real objects (Jiang et al., 2024). More recent methods map visual embeddings into linguistic space, projecting features onto language vocabularies (Neo et al., 2024) or showing the late emergence of visual signals in LLM backbones (Venhoff et al., 2025a).

In contrast, these studies primarily analyze patterns, interventions, or probing correlations, but do not directly track how multimodal fine-tuning restructures the backbone's internal features. Our work addresses this gap by providing a mechanistic perspective.

## 3 PRELIMINARIES

### 3.1 VISION–LANGUAGE MODELS

A vision–language model (VLM) consists of a visual encoder $f_V$, a pretrained language model $f_{LM}$, and a trainable projector $P$. The visual encoder (e.g., a ViT (Radford et al., 2021)) extracts image patch embeddings $V = f_V(x) = [v_1, \ldots, v_{N_V}]$, which the projector maps into token space $\tilde{V} = P(V)$. These projected image tokens are concatenated with tokenized text embeddings $T = [t_1, \ldots, t_{N_T}]$ to form the multimodal sequence $X = [\tilde{v}_1, \ldots, \tilde{v}_{N_V}, t_1, \ldots, t_{N_T}]$. Alignment between modalities is achieved through *visual instruction tuning*, where image–text pairs fine-tune the backbone to follow multimodal instructions. The language model processes $X$ through transformer layers of multi-head self-attention and feed-forward networks. For each head $h$, attention is computed as

$$\text{Attn}(Q, K, V) = \text{Softmax}\left(\frac{QK^\top}{\sqrt{d_h}} + M\right)V, \tag{1}$$

where $M$ is the causal mask preventing attention to future tokens. The outputs of all heads are concatenated and projected into the hidden dimension, and mapped through the unembedding matrix to predict next tokens. For our experiments, we adopt LLaVA-More (Cocchi et al., 2025), which extends LLaVA framework (Liu et al., 2023b; 2024) by integrating recent language models and diverse visual backbones; specifically, we use the variant combining the CLIP ViT-Large-Patch14–336 encoder with a LLaMA-3.1-8B language model backbone (Grattafiori et al., 2024).

### 3.2 SPARSE AUTOENCODERS (SAEs)

Sparse Autoencoders (SAEs) learn a dictionary of features that approximate hidden states as sparse linear combinations of interpretable directions. mitigating superposition where many features overlap in the same dimensions (Bricken et al., 2023; Cunningham et al., 2023). Formally, a vanilla SAE encodes $x \in \mathbb{R}^D$ into

$$f(x) = \text{ReLU}(W_{enc}x + b_{enc}), \quad \hat{x} = W_{dec}f(x) + b_{dec},$$

with $W_{enc} \in \mathbb{R}^{F \times D}$, $b_{enc} \in \mathbb{R}^F$, $W_{dec} \in \mathbb{R}^{D \times F}$, and $b_{dec} \in \mathbb{R}^D$. Training minimizes

$$\mathcal{L} = \|x - \hat{x}\|_2^2 + \lambda \sum_{i=1}^{F} |f_i(x)|,$$

combining reconstruction with an $L_1$ sparsity penalty. Here, decoder columns $(W_{dec})_{:,i}$ define the direction of each feature in input space, while encoder rows $(W_{enc})_{i,:}$ act as detectors that determine when a feature is present. Variants such as Top-$K$ SAEs (Gao et al., 2024) further sharpen this tradeoff by enforcing hard sparsity, improving interpretability and reducing feature co-adaptation.

SAEs have been widely applied to uncover monosemantic features and offer a practical lens on model internals, enabling analyses that range from probing knowledge to tracing safety-relevant behaviors (Bricken et al., 2023; Cunningham et al., 2023). They are not, however, a complete decomposition: interpretability can vary across runs and training setups, and recent work suggests their practical utility may be more limited in some settings (Templeton et al., 2024; Kantamneni et al., 2025). Even so, SAEs have proven particularly effective for *model diffing*, where they make it possible to track how features shift across training stages and to surface subtle but behaviorally important dynamics—a direction we expand on in the next subsection (Bricken et al., 2024; Minder et al., 2025b).

### 3.3 STAGE-WISE MODEL DIFFING

A recent line of work in model diffing has introduced *stage-wise model diffing* (Bricken et al., 2024), which extends SAE analysis across training stages by re-training dictionaries on activations from successive checkpoints while keeping feature indices aligned. This makes it possible to compare whether units are preserved, rotated, or repurposed during adaptation. Applied to controlled fine-tuning trajectories, it disentangles changes due to model updates from dataset shifts and highlights features that drive adaptation. Prior work has shown that stage-wise diffing uncovers fine-grained dynamics, including sleeper-agent features that remain dormant in pretraining but activate once safety constraints are lifted (Hubinger et al., 2024). Compared to crosscoder-based methods (Lindsey et al., 2024), it provides finer resolution at the feature level, though it remains limited to aligned checkpoints of the same architecture.

## 4 STAGE-WISE MODEL DIFFING FOR MULTIMODAL ADAPTATION

**Overview.** We aim to understand how multimodal fine-tuning reshapes model representations, using spatial reasoning as a case study of a distinctly multimodal task that integrates both visual and linguistic cues. To this end, we take inspiration from stage-wise diffing 3.3, employing sparse autoencoders (SAEs) as a feature-level lens to track how internal directions shift when a pretrained language backbone is exposed to visual inputs. Our pipeline has three stages. First, we fine-tune SAEs on multimodal activations to obtain a feature dictionary aligned with the vision–language space. Second, we isolate features that prefer visual tokens and undergo substantial geometric rotation, indicating that they have been repurposed by multimodal training. Third, we probe for spatial reasoning by contrasting generic VQA with spatial queries and keeping only features that increase under the shift while remaining active under neutral instructions, ensuring they are not driven by lexical artifacts. In this way, we reduce the original pool of over one million features to a compact set of candidates plausibly recruited for spatial reasoning tasks.

### 4.1 ADAPTING LANGUAGE DICTIONARIES TO VISION-LANGUAGE SPACE

We start by adapting sparse autoencoders (SAEs) trained on the Llama 3.1 8B backbone to the hidden states of LLAVA-MORE (Llama 3.1 8B backbone) (Cocchi et al., 2025). We use 50k image–question pairs from the VQAv2 dataset (Goyal et al., 2017), a widely used VQA benchmark of images and open-ended questions. Each SAE is attached to the output of a transformer block and trained on cached activations from these samples. Images are represented by 575 consecutive visual tokens, and questions by variable-length text sequences; this separation allows token-type–specific masking.

We initialize SAEs from the pretrained LLAMA-SCOPE release (He et al., 2024), re-instantiated as a Top-$K$ model ($k{=}50$), preserving a meaningful, interpretable basis. Since our VLM shares the same backbone, this warm-start ensures continuity with the pretrained language feature space and avoids retraining from scratch, allowing us to directly leverage millions of monosemantic features across layers. As a control, we also train SAEs from random initialization under identical conditions. Training uses Adam with a layer-scaled learning rate, and cached activations are processed in padded mini-batches. To disentangle modality-specific contributions, we consider four regimes: (i) full sequence, (ii) image-only, using only the visual-token span, (iii) text-only, using only the non-visual span, and (iv) random initialization. In all cases, the SAE receives the full hidden state sequence, but masking controls which token spans contribute to the training signal.

We evaluate reconstruction quality using the fraction of variance unexplained (FVU) and report sparsity to verify code selectivity. Evaluation is performed on a held-out split. Figure 1 shows FVU as a function of tokens seen across layers and masking regimes. Text-only SAEs converge rapidly, while image-only and full-token regimes converge more slowly to higher error, reflecting the mismatch between projector embeddings and the LLM basis. Random initialization performs worst, underscoring the importance of starting from a pretrained language dictionary. These findings establish text-only SAEs as a reliable reconstruction baseline, which we later use for model diffing.

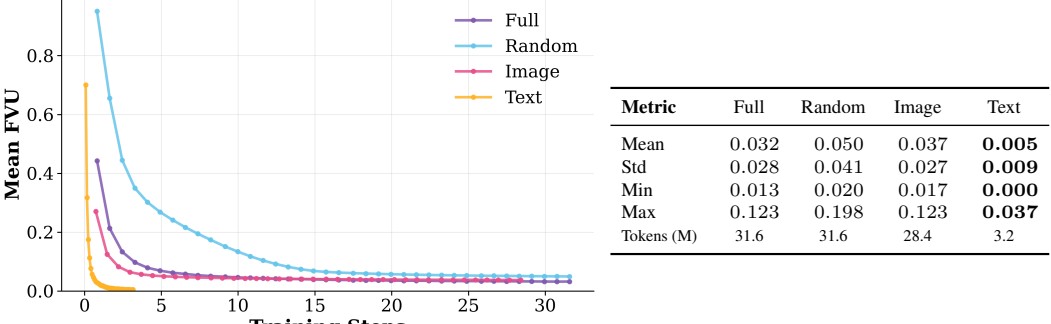

| Metric | Full | Random | Image | Text |
|--------|------|--------|-------|------|
| Mean | 0.032 | 0.050 | 0.037 | **0.005** |
| Std | 0.028 | 0.041 | 0.027 | **0.009** |
| Min | 0.013 | 0.020 | 0.017 | **0.000** |
| Max | 0.123 | 0.198 | 0.123 | **0.037** |
| Tokens (M) | 31.6 | 31.6 | 28.4 | 3.2 |

Figure 1: **SAE adaptation on LLAVA-MORE.** Left: Mean fraction of variance unexplained (FVU) across layers on the validation set. Right: Summary statistics of FVU values on the validation set, with decimal alignment; the lowest mean is highlighted in **bold**.

**Implications for stage-wise model diffing.** Stage-wise diffing assumes that fine-tuning induces *localized* (feature-level) changes rather than wholesale rotations. Prior work reports that image-token representations in early layers exhibit higher reconstruction error than text tokens, indicating a distributional gap between projector outputs and the LLM basis (Venhoff et al., 2025b). Consistent with this, our decoder–cosine analysis (Appx.Fig.5) shows that *text-only* SAEs remain highly aligned to the base LLM dictionary across layers, whereas *image-only* and *full sequence* SAEs undergo large rotations in shallow layers and only align in later layers. We also note that text-only SAEs begin with slightly higher error in the very first layers but adapt extremely quickly, converging to near-zero reconstruction. In contrast, image and full-sequence SAEs plateau at higher error, highlighting the instability of projector-driven spans (see Appx.Fig.6). We therefore focus stage-wise diffing on text-only SAEs, where alignment is stable and feature-level identifiability is more plausible.

## 4.2 Identifying Adapted Features

We aim to isolate SAE features that (i) undergo geometric reorientation after multimodal adaptation and (ii) show a clear *modality preference* for vision input. Such features are the most informative for model diffing and subsequent causal analysis. To identify them, we rely on two signals:

**1. Geometric reorientation (decoder cosine).** To test if $f$ has been *repurposed* by multimodal fine-tuning, we compare its decoder direction before and after adaptation. Let $W_{\text{dec},f}^{\text{LLM}}$ be the base SAE decoder vector and $W_{\text{dec},f}^{\text{VLM}}$ the corresponding vector in the VLM-adapted SAE. We compute

$$c_f \;=\; \cos\!\big(W_{\text{dec},f}^{\text{LLM}}, W_{\text{dec},f}^{\text{VLM}}\big).$$

High $c_f$ means the semantic direction of $f$ stayed aligned with the original language dictionary; low $c_f$ indicates a substantial rotation, consistent with a reallocation of $f$ to encode new multimodal structure. We use decoder vectors rather than encoder parameters because decoder directions more directly index the feature's semantics.

**2. Modality preference (visual energy).** Given the sparsity of SAE activations, we score each feature $f$ by its mean squared activation under vision inputs,

$$E_v(f) \;=\; \mathbb{E}_{\text{vision}}\big[h_f^2\big],$$

measured on VQA runs of the VLM. Since nearly half of features have $E_v = 0$, a simple cutoff $E_v > \epsilon$ suffices to discard inactive directions and retain those that carry visual signal.

**Selection Procedure** We define adapted features as those that meet both criteria: $E_v > \epsilon$, ensuring reliable visual responsiveness, and a cosine similarity $c_f$ in the bottom $p_{cos} = 25\%$, indicating strong decoder rotation. Applying these filters jointly yields a globally defined set comprising about 5% of all features. The joint distribution of $E_v$ and $c_f$ is shown in Fig. 2, with the selected subset highlighted in pink. Details on threshold choices, together with per-layer counts and mean cosine similarities, are provided in Appx. Fig. 7a and Appx. 7b.

## 4.3 Case Study: Identifying Spatial Reasoning Features

We identify spatial features using two signals: (i) recruitment under a shift to spatial queries, and (ii) persistence under neutral prompts that rule out lexical artifacts.

**Datasets.** Our analysis uses two evaluation sets from VQAv2. The baseline is the full validation split, denoted $\mathcal{D}_{\text{base}}$. To induce a targeted shift, we construct a spatial subset $\mathcal{D}_{\text{sp}}$ by filtering questions that contain spatial cues (e.g., *left/right/above/behind*). This contrast tests whether some SAE features are selectively recruited under spatial reasoning.

**1. Distribution shift** Let $h_f(x_t) \geq 0$ denote the activation of feature $f$ on token $t$ of input $x$. For a dataset $\mathcal{D}$, the firing frequency of $f$ is

$$p_f(\mathcal{D}) \;=\; \frac{1}{n(\mathcal{D})} \sum_{x \in \mathcal{D}} \sum_t \mathbf{1}\{h_f(x_t) > 0\},$$

where $n(\mathcal{D})$ is the total number of tokens. We compute this measure for the base split $\mathcal{D}_{\text{base}}$ and a spatial split $\mathcal{D}_{\text{sp}}$, and evaluate each feature using the frequency gap $\Delta p_f = p_f(\mathcal{D}_{\text{sp}}) - p_f(\mathcal{D}_{\text{base}})$

alongside its odds ratio $\text{OR}_f$. Features with meaningful $\Delta p_f$ and $\text{OR}_f$ are flagged as spatial *candidates* in Fig. 2. Further details, including firing-frequency and scatter-plot visualizations for both splits, are provided in Appx. A.5.

**2. Filtering lexical artifacts.** To rule out prompt-lexical effects, we replace the original questions in each top-activating sample with neutral spatial prompts such as *"Describe the positions of objects in the image."*. Features that continue firing under these generic instructions are preserved as genuinely image-grounded, while those that fail to activate are discarded. This ensures that the surviving units reflect spatial reasoning rather than memorized lexical cues.

From these filtered candidates, we retain only those also in the adapted set $\mathcal{A}$ (Sec. 4.2), ensuring they reorient under multimodal fine-tuning and respond to spatial shifts. The surviving features are shown in Fig. 2 (blue). A subset, marked with red crosses, is further analyzed via automated interpretation, attribution patching, and ablations (Sec. 5.1, 5.2).

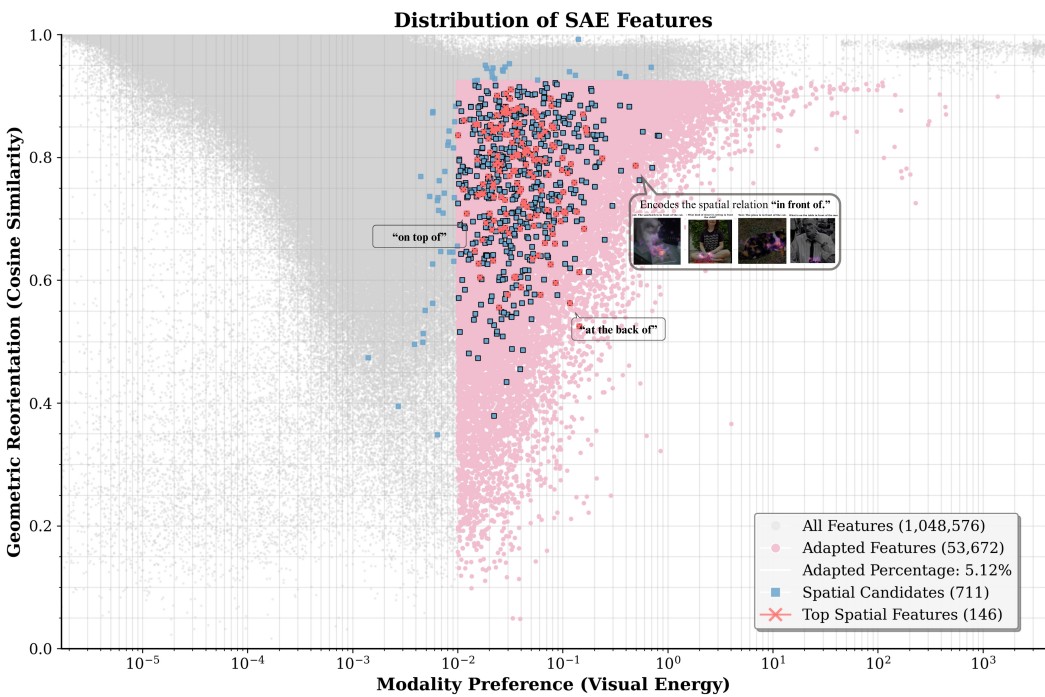

Figure 2: **Distribution of SAE features by visual energy and cosine similarity.** All features are shown in gray; adapted features are highlighted in pink. Spatial candidates are marked with blue squares, and the subset used for downstream analysis is shown as red crosses.

**Extension to OCR-style prompts.** While our primary case study focuses on spatial reasoning, the same feature-selection procedure can be applied to other visually grounded skills. As a second case study, we analyze features associated with visual text recognition by contrasting OCR-style prompts (e.g., "What does the sign say?") with generic VQA questions. We construct an OCR-focused split by filtering VQAv2 images that contain legible embedded text and computing feature firing frequencies under the same distribution-shift statistics used for spatial queries. This reveals a compact subset of adapted units whose activations increase on OCR prompts and remain non-zero under neutral image descriptions, indicating that they are tied to image-grounded text rather than specific lexical patterns. Additional qualitative examples and follow-up analyses are provided in Appx. A.6, where these OCR-selective features are shown to align with regions containing characters and words and to be supported by a small number of recurring mid-layer heads.

# 5 EXPERIMENTS

## 5.1 AUTO-INTERP AND PRELIMINARY INSPECTION

As an initial step toward understanding the selected features, we carried out a preliminary inspection using an automated interpretation pipeline. For each feature, we collect its top-activating samples from two sources: general VQA questions from VQAv2 (not restricted to spatial reasoning) and the Visual Spatial Reasoning (VSR) dataset (Liu et al., 2023a), which is inherently spatial. This pairing allows us to check whether the same underlying meaning emerges consistently across both settings (Fig. 3). A subset of the combined samples are then passed to the `gpt-4o-mini` (OpenAI, 2024) API, which proposes a concise one-sentence description for each feature and assigns an interpretability confidence score based on F1 from a validation classification task. The resulting outputs are stored together with the selection metrics from Sec. 4.2, and are lightly reviewed by hand, so that the retained set reflects both automatic labeling and human verification (see App. B.1 for additional examples and scoring details)..

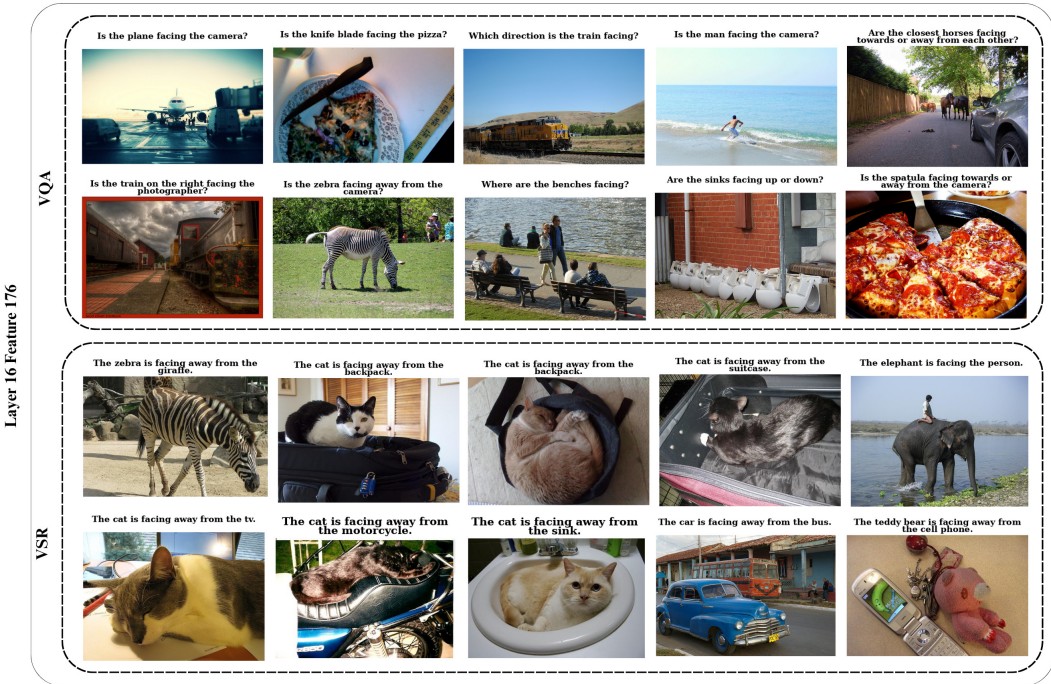

Figure 3: **Auto-Interp example (Layer 16, Feature 176).** Top VQA and VSR samples both highlight *facing direction*, with activation on objects described as facing toward, away, or relative to another.

## 5.2 ATTRIBUTION PATCHING TO IDENTIFY SPATIAL HEADS

**Method.** Attribution patching (Nanda, 2023) is a scalable alternative to activation patching (Zhang & Nanda, 2024), which measures causal effects by replacing activations with counterfactuals. While activation patching requires a separate forward pass per intervention, attribution patching uses a gradient-based linear approximation to estimate interventions with two forward and one backward pass. This makes it practical to probe attribution scores across layers and heads in MLLMs.

We adapt attribution patching to identify which attention heads drive spatially selective SAE features. For a target feature $f$ at layer $L$, we define a scalar objective by projecting the layer-$L$ activations onto the SAE decoder vector. Gradients of this objective w.r.t. upstream query/key activations indicate how strongly each attention head contributes to $f$. We compare two runs:

- **Clean run:** the original image–text input.

- **Corrupt run:** the same input, but with layer-0 visual token embeddings replaced by a *mean embedding* computed over many VQA samples. This corruption preserves plausible distributional statistics while deliberately suppressing spatial information.

We then compute two attribution variants, differing in whether the perturbation direction is taken from the corrupted or the clean representation:

$$\text{Method A:} \quad (\text{corr} - \text{clean}) \cdot \nabla_{\text{clean}},$$
$$\text{Method B:} \quad (\text{clean} - \text{corr}) \cdot \nabla_{\text{corr}}.$$

Method A measures how strongly the clean gradients indicate that ablating spatial detail affects the feature, whereas Method B measures how strongly the corrupted gradients indicate that retaining spatial detail matters. In both cases, we obtain per-layer and per-head attribution scores, averaged over the top-$k$ VQA samples that most strongly activate $f$.

**Results.**  Across the spatially selective features we examined, attribution patching with both methods reveals consistent trends. Layer-wise attribution curves typically peak in middle layers, consistent with the emergence of spatial features in Sec. 4.3 (Appx. Fig. 12). At the head level, both methods generally highlight a small subset of heads with notably high scores, and the top heads identified are often consistent across the two attribution methods (Appx. Fig. 13). This suggests that spatial information is mediated by a specialized group of heads rather than being spread uniformly across the model.

To illustrate the effect of attribution patching on individual features, Appx. Fig. 14 provides detailed examples. In each case, attribution scores isolate a handful of heads, and qualitative maps confirm that high-scoring heads focus on regions consistent with the queried relation (e.g., "on top of," "behind"), whereas low-scoring heads fail to do so. These head-level overlays can also be used to (i) improve the confidence of automated feature interpretation by coupling sample activations with attention visualizations, and (ii) examine failure cases by checking whether the top spatial features and heads attend to valid regions in misclassified samples. Interestingly, when we look across multiple related spatial features together, we find that some of the same heads recur across related spatial relations. Fig 4 illustrates this pattern. In the top row, L13H1 attends to semantically relevant regions across queries. As a control, the middle row shows that bottom-ranked heads on the same samples fail to localize meaningfully. The bottom row further confirms that irrelevant queries do not trigger spurious activation. More generally, these same heads also attend to meaningful regions such as salient objects or attributes under custom prompts (Appx. Fig. 16), underscoring that attribution patching identifies a set of heads that reliably carry spatial–semantic signal.

## 5.3 ABLATION STUDY

We test whether adapted SAE features are *causally involved* in spatial reasoning by ablating them during inference and measuring performance on VSR (Liu et al., 2023a), a dataset of text–image pairs spanning dozens of spatial relations, and on a *Yes/No* subset of VQAv2 (general). Each feature is evaluated on a *relation-specific subset* of VSR constructed from its top-activating samples, so that the ablation directly targets the relation it most strongly encodes. To ablate a target feature $f$ at layer $L$, we orthogonally remove its decoder direction $v$ (unit norm) from the residual stream at *text* token positions, leaving image tokens unchanged:

$$y \leftarrow y - (y^\top v)\, v.$$

**Evaluation metrics.**  We report: (i) accuracy drop on VSR ($\Delta$VSR Acc; $\downarrow$ is worse), (ii) accuracy drop on VQA ($\Delta$VQA Acc), (iii) accuracy drop from ablating same-layer random features ($\Delta$Ctrl), and (iv) odds ratio under the spatial distribution shift (VSR OR; $\uparrow$ is better). All runs use identical cached indices, and results are averaged over seeds.

**Interpretation.**  Ablating the top spatial features lowers VSR accuracy by 9–16 points on average while leaving general VQA nearly unchanged ($\leq 1$ pp), indicating that these directions are functionally used for spatial reasoning rather than general behavior. This shows that probing or switching off a single feature can selectively disable spatial reasoning without harming overall ability. High odds ratios further show selective recruitment under spatial prompts. Random-feature controls yield effects near zero or inconsistent in sign, supporting specificity. Full per-feature results, probability deltas, and seed-wise summaries are reported in Appx. D.

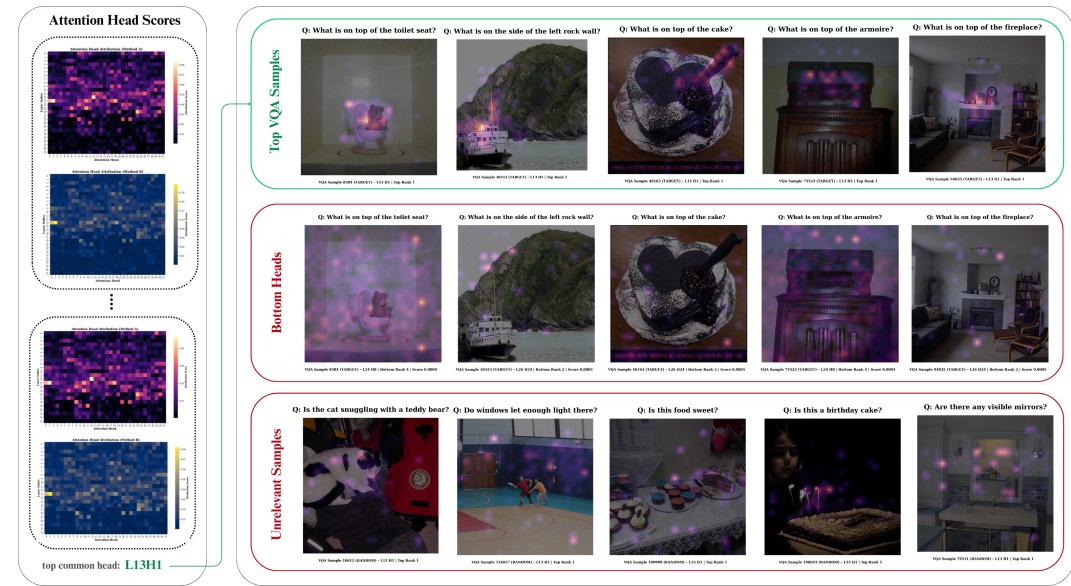

Figure 4: Attribution patching across related spatial features. Top: recurring top-scoring head (L13H1) localizes to relevant regions in queries about "on top of" relations. Middle: bottom-ranked heads on the same samples fail to capture spatial structure. Bottom: unrelated queries confirm that the top head does not spuriously activate.

| Layer | Feature | ΔVSR Acc (↓) | ΔVQA Acc (↓) | ΔCtrl | VSR Relations | VSR OR (↑) |
|-------|---------|--------------|--------------|-------|---------------|------------|
| 7 | 15870 | -15.54 | -0.10 | -0.88 | above | 4.32 |
| 11 | 27061 | -12.77 | -0.40 | 0.00 | across from | 8.03 |
| 9 | 15404 | -11.19 | -0.80 | 1.08 | below | 5.60 |
| 14 | 17873 | -10.21 | -0.30 | -1.71 | at the right side of | 7.17 |
| 12 | 23874 | -9.05 | -0.40 | -0.95 | left of | 9.10 |
| 18 | 29948 | -7.98 | -0.30 | 0.00 | beside | 8.36 |

Table 1: **Top ablated SAE features** ranked by VSR accuracy drop. Columns 2–4 show accuracy drops on VSR, VQA, and random-feature controls; the final column gives odds ratio (VSR OR) as a measure of selective recruitment. Large ΔVSR Acc with small ΔVQA Acc indicates spatial specificity, while near-zero ΔCtrl confirms robustness.

## 6 LIMITATIONS

Our analyses indicate spatial selectivity, but more detailed ablation and steering studies are needed to fully validate causality. Moreover, our experiments are limited to a single model (LLaVA-More with a LLaMA-3.1-8B backbone); applying the method to other backbones and larger corpora will be key to assessing generality.

## 7 CONCLUSION

We set out to understand how a pretrained language backbone learns to "see" under multimodal fine-tuning. By extending stage-wise model diffing to the vision–language setting, we isolated vision-preferring features that undergo strong rotations during training, showed that a subset reliably encodes spatial relations, and traced their causal drivers to a small number of mid-layer attention heads. These results show that multimodal adaptation is structured and interpretable as it can be localized, probed, and explained at the feature level. Beyond spatial reasoning, our methodology offers a general framework for uncovering when and where new capabilities emerge in large models, showing that multimodal adaptation follows structured patterns rather than diffuse changes. We view

this work as an early step toward a mechanistic science of multimodal training, where models can be interpreted both in terms of their outputs and the internal features that support them.

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

# A  APPENDIX

## A.1  GEOMETRY DIVERGENCE: DECODER COSINE TRENDS

To quantify how SAE feature geometry shifts across training regimes, we track cosine similarity between decoder directions from SAEs trained on different input types. Fig 5 shows that text-only SAEs remain closely aligned across layers, while image-only and full-sequence SAEs diverge in early layers before realigning deeper in the model. Randomly initialized SAEs stay largely uncorrelated, confirming the stability of the observed trends.

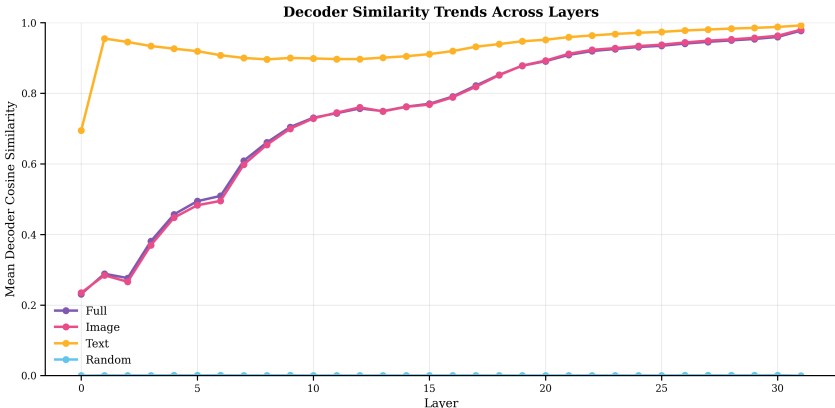

Figure 5: **Decoder cosine similarity vs. layer (LLM SAE vs. VLM SAE).** Text-only stays highly aligned across layers; image-only and full-sequence rotate in shallow layers and align later; random remains near zero. Higher cosine indicates closer alignment of SAE decoder directions.

## A.2  PER-LAYER FVU TRAJECTORIES

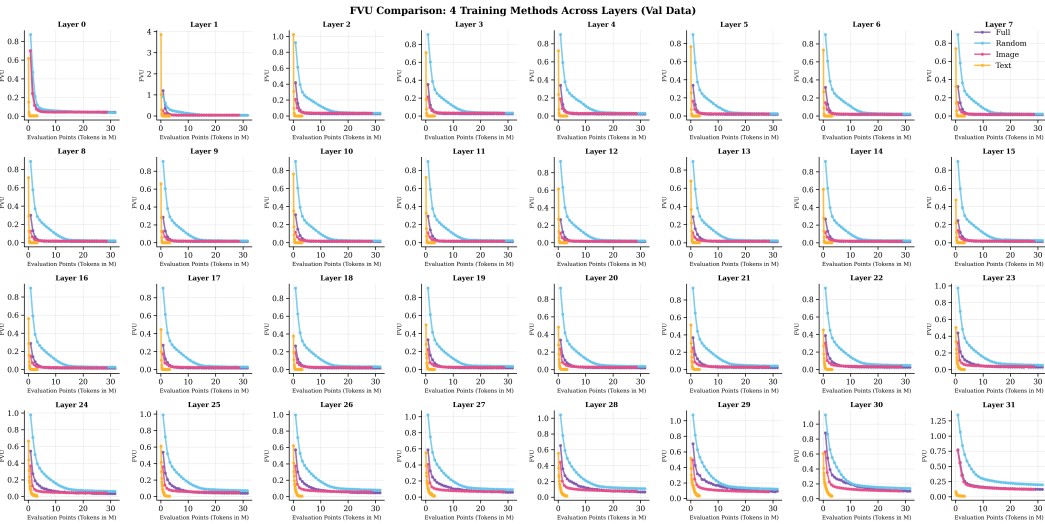

Figure 6: **Per-layer FVU across regimes.** Each panel shows the convergence of SAEs trained with different masking regimes for a specific layer. Text-only SAEs begin with slightly higher error in the shallowest layers but adapt almost immediately to near-zero reconstruction. Image-only and full-sequence SAEs converge more slowly and plateau at higher error, while random initialization performs worst throughout. This confirms that projector-driven spans remain off-distribution in early layers and only align with the LLM basis in later layers.

## A.3   Per-Layer Statistics

Fig 7 shows that adapted features cluster in mid layers and taper in deeper blocks. Their decoder directions remain less aligned to the base dictionary than the overall pool, confirming stronger rotations under multimodal fine-tuning.

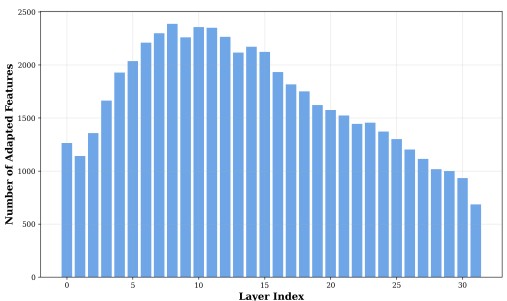
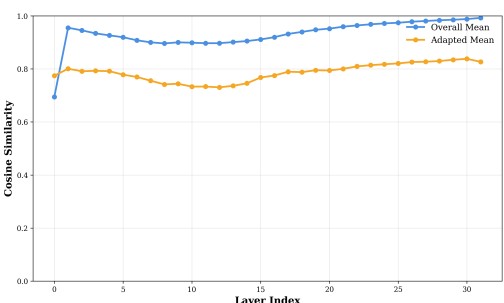

(a) **Adapted features per layer.** Most concentrate in mid layers, tapering in deeper blocks.

(b) **Decoder cosine by layer.** Adapted features remain less aligned to the base dictionary than the overall pool.

Figure 7: **Per-layer statistics of adapted features.** (a) Distribution of adapted feature counts across depth. (b) Mean decoder cosine similarity for adapted features vs. the overall pool.

## A.4   Threshold Sweep for Feature Selection

To ensure that our choice of thresholds is robust, we sweep over the cosine percentile cutoff ($p_{\cos}$) and visual energy threshold ($\epsilon$). Fig. 8 reports three metrics: (i) total number of selected features, (ii) Jaccard overlap with the baseline adapted set, and (iii) per-layer count correlation. The results show a broad stable region around $\epsilon \approx 10^{-3}$ and $p_{\cos} \approx 25\%$, which yields a compact yet consistent set of adapted features. We adopt this operating point (white circle) for all downstream analyses.

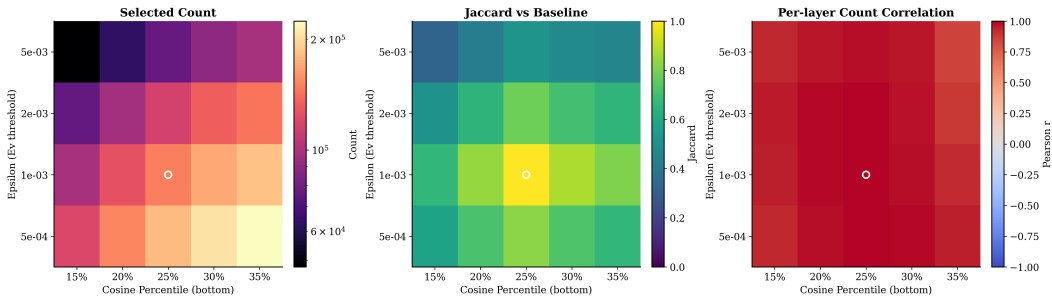

Figure 8: **Threshold sweep for feature selection.** Left: feature counts increase smoothly with more lenient thresholds. Middle: Jaccard overlap with the baseline peaks near the chosen point. Right: per-layer counts remain highly correlated across thresholds. The white circle marks the adopted operating point.

The visual-energy statistic $E_v$ is computed under a text-only mask, since our SAEs are text-only. As a result, most features have $E_v = 0$, so requiring $\epsilon > 0$ acts as a strong filter. When cross-checking with downstream spatial tasks, we find that features with very low $E_v$ rarely contribute meaningfully: they tend to cluster in shallow layers, show low spatial hit rates, and often appear polysemantic on inspection. In contrast, those that pass the $\epsilon$ cutoff carry a cleaner visual signal and align more consistently with spatially selective units in downstream evaluations, suggesting that the thresholded set captures genuinely vision-grounded features.

## A.5 Distribution-shift visualizations

To complement the main-text description of our feature-selection procedure, we include here the firing-frequency distributions and candidate-feature scatter plots used to identify spatial units under different prompting conditions.

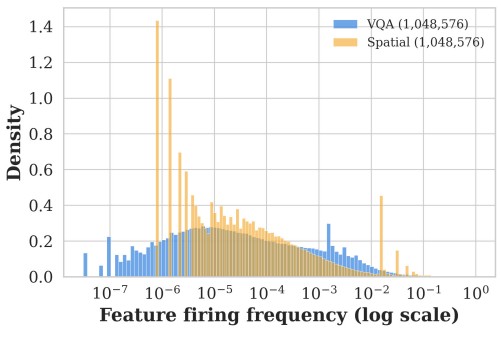

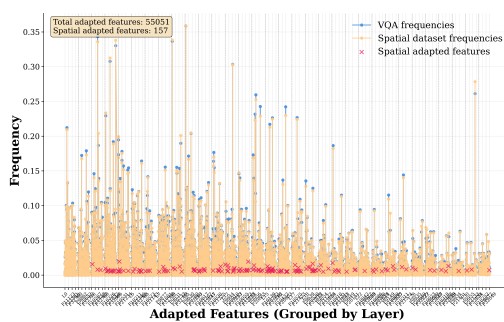

(a) Firing-frequency distributions for $\mathcal{D}_{\text{base}}$ and the spatial split $\mathcal{D}_{\text{sp}}$.

(b) Spatial candidate features under both splits, with selected units highlighted.

Figure 9: **Spatial distribution shift.** Visualization of feature firing frequencies and candidate selection under the spatial vs. base splits.

## A.6 OCR feature visualizations

We also apply our distribution-shift procedure to OCR-style prompts (e.g., "What does the sign say?"). Fig. 10 shows that OCR-selective features cluster within the same adapted region as the spatial subset, indicating that multimodal fine-tuning concentrates visually grounded capabilities into a compact envelope of feature space.

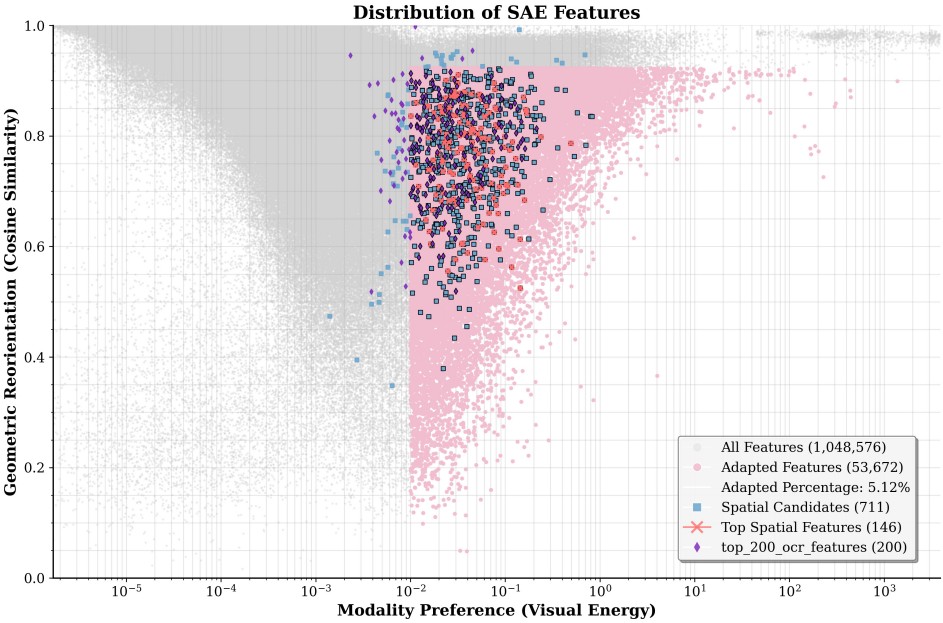

Figure 10: **Distribution of OCR features.** Top OCR candidates (purple) cluster among adapted units (pink), paralleling the spatial subset (blue).

# B  ADDITIONAL AUTO-INTERP EXAMPLES

In the main text (Sec. 5.1), we showed examples of adapted features using our automated interpretation pipeline. We include two further examples here. In both cases, the top-activating samples agree across VQA and VSR, and the interpretations are consistent and monosemantic.

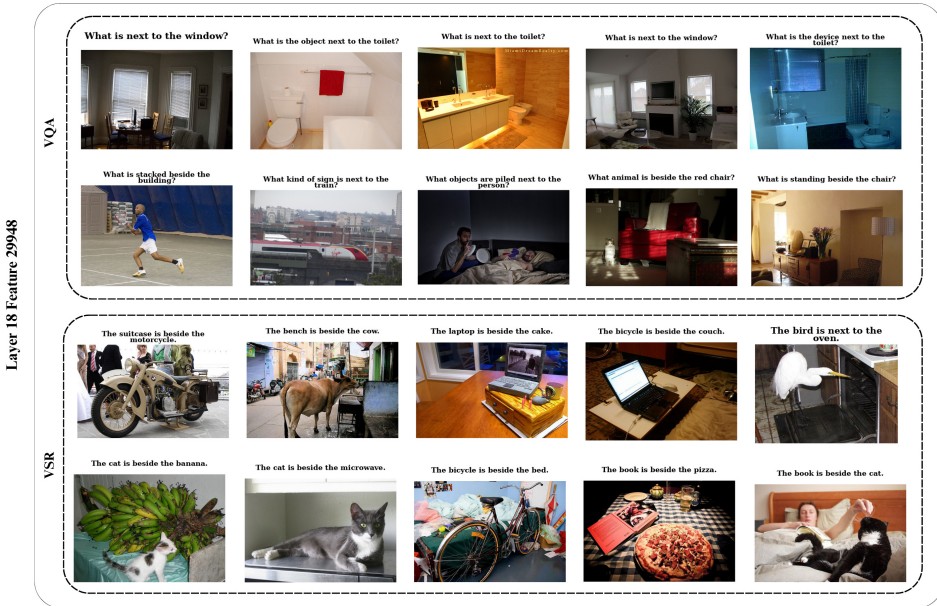

description: "this neuron activates for beside or next to relations between objects.", F1: 0.8

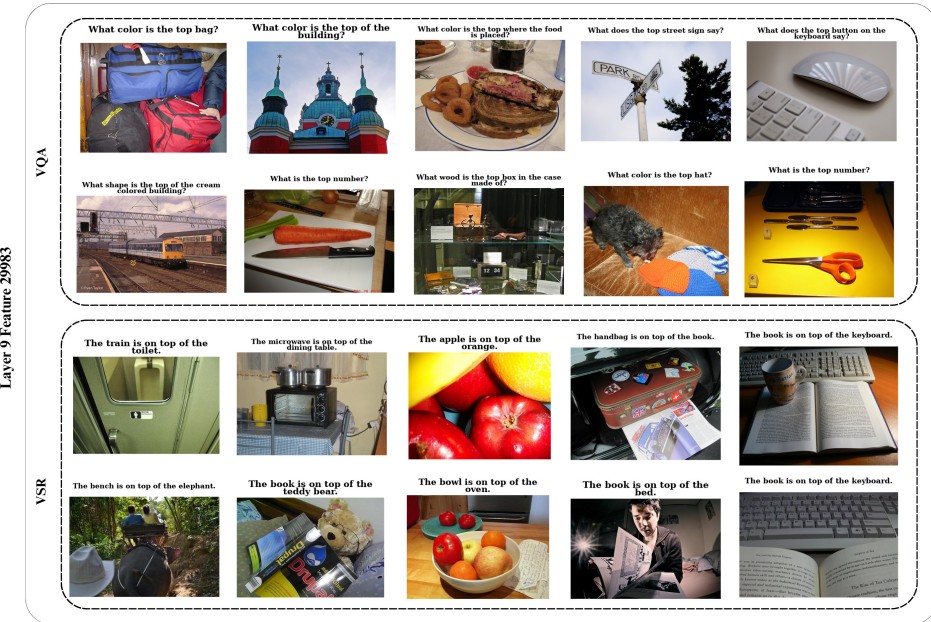

description: "this neuron activates for top of relations.", F1: 0.7

Figure 11: **Additional Auto-Interp examples.** Top-activating VQA and VSR samples for two adapted features, showing consistent spatial relations.

## B.1 AUTO-INTERPRETATION AND SCORING PIPELINE

We evaluate interpretability using an automated feature-description pipeline with two variants: *RAW* (image+text) and *OVERLAY* (image+text+top-head heatmaps). For each feature $f$: 1. Select up to $k=5$ top-activating samples (deduped across VQA/VQA-spatial/VSR). 2. Call the API once to generate a single concise description. 3. Validate using held-out positive samples and random VQA negatives (two short rounds). 4. Compute F1 as a lightweight proxy for description confidence.

Outputs are stored per feature as JSON (description, examples, classification results). Adding overlays improves interpretability, with early results showing a typical gain of about $+0.2$ F1.

---

**PROMPT A: Description (RAW / OVERLAY)**

**System:** You are analyzing individual neurons using their top-activating samples (image+text; OVERLAY also includes attention heatmaps).
**Task:** Produce *one* short, lower-case sentence completing: "this neuron activates for . . . ".
**Guidelines:** Base it on consistent patterns supported by image (+ overlays) and text; be specific; no hedging.
**Return:** { "description": "one concise sentence" }.

---

**PROMPT B: Validation (F1)**

**System:** You are validating a neuron description against short examples (image+text; OVERLAY adds heatmaps).
**Task:** For each sample, output 1 if it reasonably matches the description; else 0.
**Return:** { "classifications": [0/1, . . . ] }.

---

## C ATTRIBUTION PATCHING ADDITIONAL EXPERIMENTS RESULTS

### C.1 AGGREGATED ATTRIBUTION RESULTS

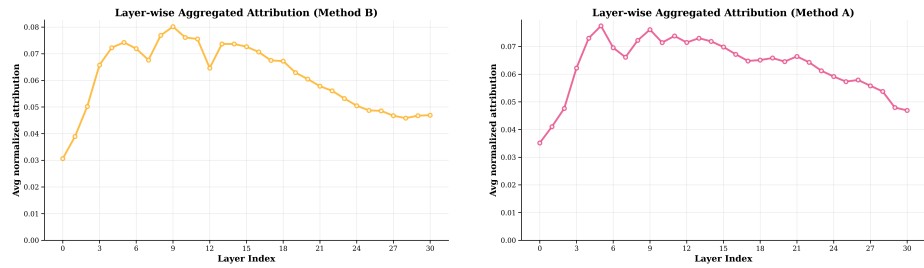

Figure 12: Layer-wise aggregated attribution curves for Method B (left) and Method A (right). Both peak in around middle layers, consistent with the emergence of spatial features.

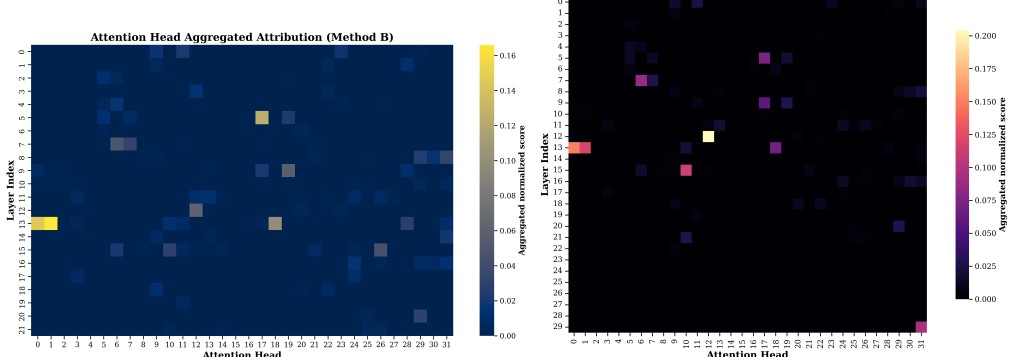

Figure 13: Attention head aggregated attribution maps for Method B (left) and Method A (right). Both highlight a similar set of specialized heads with high attribution scores.

## C.2 Per-Feature Panels with Top Heads

For individual spatial features, we show (i) per-layer/head attribution maps (Methods A and B) and (ii) attention overlays from the strongest heads on the feature's top-activating samples across both VSR and VQA datasets.

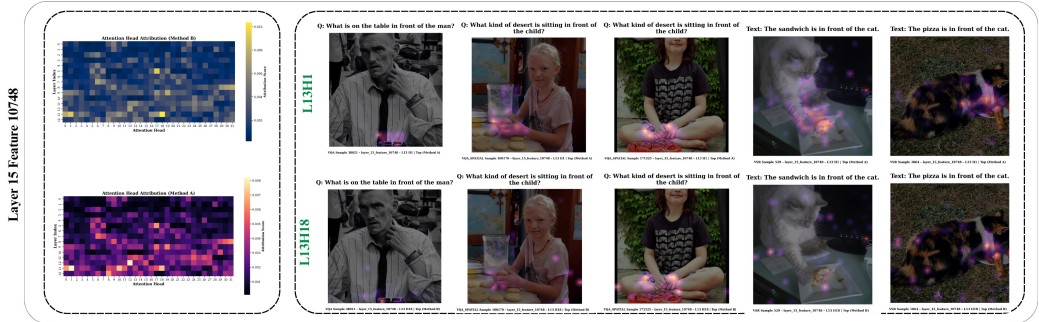

(a) **Layer 15, Feature 10748.** VSR Relation: "in front of." Top heads (Method A): `L13H1`, `L12H12`, `L13H18`. Top heads (Method B): `L13H18`, `L5H17`, `L13H1`. *Overlap*: `L13H1`, `L13H18`. Attention overlays are shown on the top-activating samples across VSR and VQA.

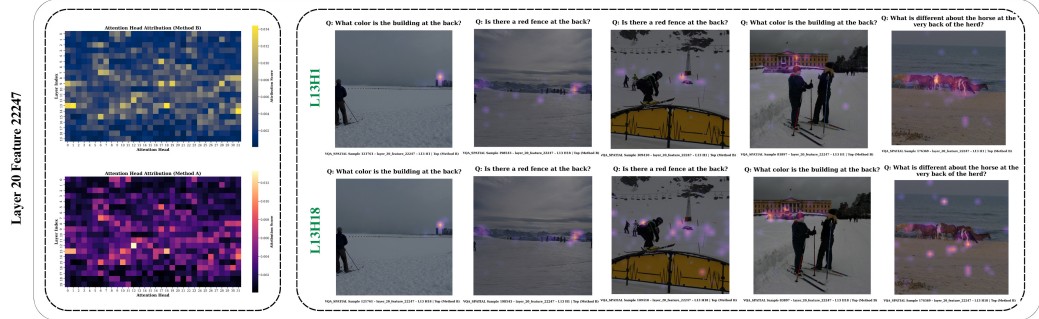

(b) **Layer 20, Feature 22247.** VSR Relation: "at the back of." Top heads (Method A): `L12H12`, `L13H18`, `L13H1`. Top heads (Method B): `L13H1`, `L13H18`, `L14H31`. *Overlap*: `L13H1`, `L13H18`. Attention overlays are shown on the top-activating samples across VSR and VQA.

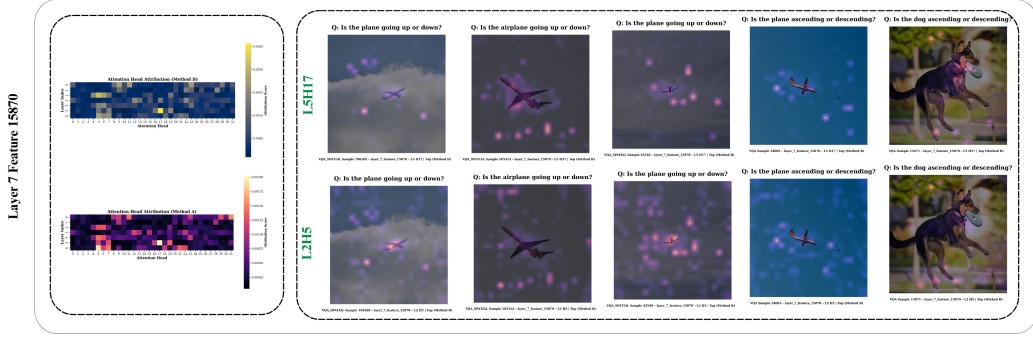

(c) **Layer 7, Feature 15870.** VSR Relation: "above." Top heads (Method A): `L5H17`, `L6H5`, `L0H31`. Top heads (Method B): `L5H17`, `L2H5`, `L2H6`. *Overlap*: `L5H17`. Attention overlays are shown on the top-activating samples across VSR and VQA.

Figure 14: **Attribution patching on individual spatial features.** Each subfigure displays aggregated head/layer attribution maps (left) and attention overlays (right) using the strongest heads on the feature's top-activating samples across both VSR and VQA.

Across these examples, the two attribution methods consistently surface overlapping heads, indicating that a small group concentrates much of the spatial signal. Method B generally produces sharper rankings and cleaner overlays, suggesting it is more reliable for identifying the causal drivers of spatial features.

## C.3  BOTTOM-RANKED HEADS AS A CONTROL

As a control, we visualize overlays from the *bottom-ranked* heads (per method, per feature). Across VSR and VQA top-activating samples, these heads generally fail to localize semantically relevant regions.

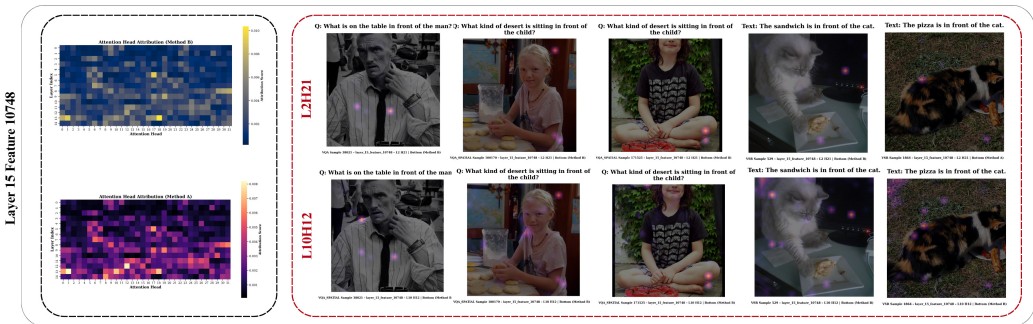

(a) **Layer 15, Feature 10748.** VSR Relation: "in front of."

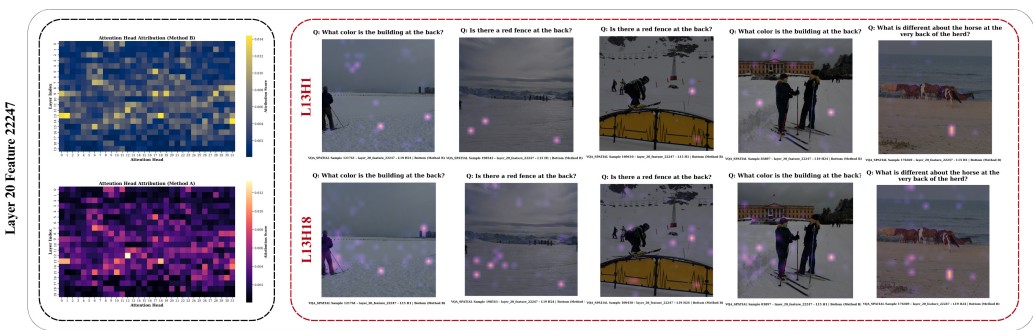

(b) **Layer 20, Feature 22247.** VSR Relation: "at the back of."

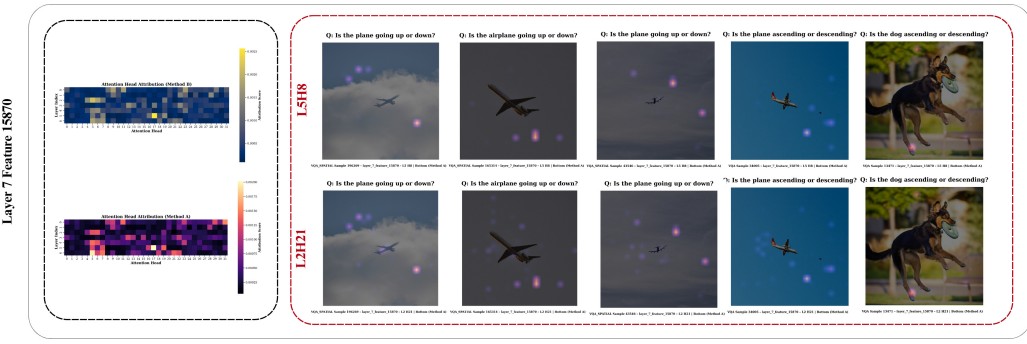

(c) **Layer 7, Feature 15870.** VSR Relation: "above."

Figure 15: **Bottom-ranked heads yield weak localization.** For each feature, we show overlays from the lowest-scoring heads under Methods A and B on the feature's top-activating samples across VSR and VQA. In contrast to Appx. Fig. 14, these heads produce diffuse or irrelevant attention.

## D  FULL ABLATION RESULTS

Table 2 reports a more detailed version of ablation results for the top SAE features. For each feature, we show average accuracy and probability drops on VSR across seeds, together with the number of evaluation samples. We also report accuracy drops on VQA, random-feature control drops ($\Delta$Ctrl), odds ratios (VSR OR), and relation-specific subsets of VSR derived from top-activating samples.

Large negative ΔVSR Acc with small ΔVQA Acc indicates spatial specificity, near-zero ΔCtrl supports robustness, and high odds ratios reflect selective recruitment under spatial prompts.

| Layer | Feature | ΔVSR Acc | ΔVSR Prob | ΔVQA Acc | ΔCtrl | VSR OR | #Samples | VSR Relations |
|-------|---------|----------|-----------|----------|-------|--------|----------|---------------|
| 11 | 27061 | -13.30 | -0.09 | -0.40 | 0.00 | 8.03 | 94 | across from |
| 12 | 23874 | -10.24 | -0.10 | -0.40 | -0.95 | 9.10 | 210 | left of |
| 18 | 29948 | -7.98 | -0.09 | -0.30 | 0.00 | 8.36 | 188 | beside |
| 23 | 4060 | -5.85 | -0.00 | -0.70 | -1.06 | 7.47 | 94 | at the back of |
| 14 | 17873 | -10.00 | -0.07 | -0.30 | -2.71 | 7.17 | 480 | at the right side of |
| 9 | 15404 | -11.19 | -0.07 | -0.80 | 1.08 | 5.60 | 277 | below |
| 7 | 6986 | -10.87 | -0.03 | -0.50 | 0.34 | 4.74 | 589 | under |
| 7 | 15870 | -15.54 | -0.09 | -0.10 | -0.88 | 4.32 | 341 | above |
| 10 | 5121 | -7.92 | -0.06 | -0.10 | 0.12 | 4.22 | 846 | above, on top of |
| 11 | 24089 | -7.68 | -0.05 | -0.60 | -0.12 | 4.18 | 846 | above, on top of |
| 12 | 13305 | -6.38 | -0.05 | -0.70 | 0.24 | 4.17 | 846 | above, on top of |

Table 2: Full ablation results for top SAE features, averaged over seeds. The number of VSR samples evaluated is shown alongside accuracy/probability drops and odds ratios.

# E

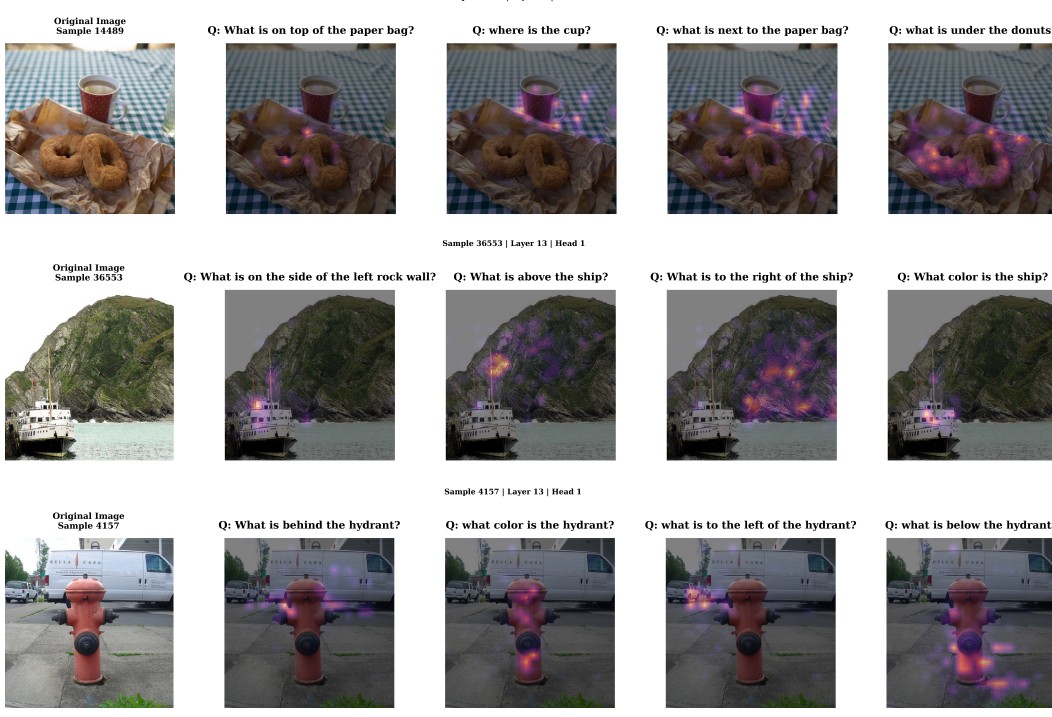

Figure 16: **Attention head visualizations across queries.** Each row shows one image with attention overlays from a single high-attribution head across multiple spatial and non-spatial custom queries. The same heads consistently focus on semantically relevant regions.

