# OpenReview forum: "Towards Understanding Multimodal Fine-Tuning: A Case Study into Spatial Features"
_ICLR.cc/2026/Conference — Submitted to ICLR 2026_

### Official Review · Reviewer_WZEM · 2025-10-29

**Soundness:** 3
**Presentation:** 3
**Contribution:** 3
**Rating:** 4
**Confidence:** 3

**Summary:**

The paper presents an empirical study of internal representation in VLMs during the visual fine-tuning phase. The authors adapt the stage-wise model diffing technique to investigate representation changes during the visual fine-tuning of a LLaMA model with a pre-trained LLaMA3.1 textual backbone. Model diffing allows to measure the directional changes in the feature representations encoded by Sparse Autoencoders (SAE) trained on sequential model checkpoints during the fine-tuning. This enables the location of changes within the model's feature representation via cosine distance measures. The authors then propose to focus on a small subset (~5%) of features which show significant change and are related to spatial information. In a subsequent analysis, the paper shows that only a small number of features in mid-layers attention heads are emerging to learn visual spatial information during the fine-tuning and concludes that the presented workflow could be utilized to gain deeper understanding of how multi-modal models learn their representations.

**Strengths:**

The paper is well written and mostly easy to follow. The presented idea is novel in its application to VMLs / MMLMs and tackles an interesting and practically significant research question. The presented results are not fully convincing (see weaknesses) , but definitely intriguing and likely to trigger followup work.

**Weaknesses:**

The main weakness of this paper is already identified by the authors in the limitation section: the entire evaluation is based on a single VLM fine-tuning, using a single LLM backbone and a single dataset. This raises server questions regarding the generality of the findings. Moreover, the entire workflow has a lot of "moving parts", e.g. adjustable parameters like training hyper-parameters of the fine-tuning, hyper-parameters of the SAE training and feature selection thresholds. The paper presents only results for a single point in this huge parameter space, leaving the question unanswered how stable the presented results (and the derived conclusions) actually are if one would change one or more of the many parameters.
While the parameter space is way to large to ask to ablate all possible combinations, the reviewer would strongly suggest to study at least some of the most obvious parameters:
* show results for more than one model
* more than one data-set
* ablate  $k$ in the SAE
* ablate the feature thresholds
* ...

Minor points:
* some figures have tiny captions (within the images) which are unreadable even at 300% zoom

**Questions:**

Q1: it is not quite clear how the SAE indices are aligned between different the different checkpoints

---

> ### Author Response · Authors · 2025-12-04
>
> Thank you for your detailed and constructive feedback. We appreciate the opportunity to clarify our design choices and have incorporated these points into the revised draft.
>
> > The evaluation covers a single model and a single dataset; the parameter space is large.
>
> We agree that our empirical scope is limited, and we now explicitly present this work as a targeted mechanistic case study rather than a general evaluation across architectures. The framework itself makes minimal assumptions — requiring only aligned checkpoints and hidden activations — so it can directly extend to newer backbones and training regimes as compute allows. Given current SAE availability, we prioritized depth of analysis over breadth of configurations.
>
> > There are many “moving parts” (SAE setup, thresholds, dataset choices, etc.); unclear robustness.
>
> While it is infeasible to exhaustively sweep the full hyperparameter space for a model at this scale, we include robustness checks for the components most likely to affect conclusions:
>
> • Feature thresholds: We now report sweeps over our adaptation and visual-energy cut-offs, showing that while exact counts vary, the layerwise structure and presence of a compact adapted subset remain stable across a broad range.
>
> • Datasets / tasks: Beyond spatial reasoning, we summarize an OCR case study using the same method, recovering a parallel group of visually grounded features. This indicates the findings are not specific to one dataset or task.
>
> • SAE configuration: We adopt a standard, well-validated SAE setup (from Llama-Scope) and keep it fixed across all regimes, ensuring differences arise from model adaptation — not changes in the interpretability tool.
>
> We have clarified these robustness aspects in the revision and emphasize extension to additional models as a primary direction for future work.
>
> > Some figure captions are too small and text is unreadable upon zoom.
>
> Thank you, we have replaced all figures with vector graphics and enlarged captions/labels for full readability under zoom.
>
> > Q1: How are SAE indices aligned across checkpoints?
>
> We follow the standard stage-wise model-diffing approach: a single SAE dictionary is first trained on the pretrained model’s activations and then fine-tuned on each checkpoint while preserving feature index ordering. Thus, feature i remains the same dictionary unit across stages, allowing us to track changes in both decoder direction and activation strength through the fine-tuning process. We have clarified this procedure in the main text for transparency.

---

### Official Review · Reviewer_9Shp · 2025-10-30

**Soundness:** 2
**Presentation:** 2
**Contribution:** 1
**Rating:** 4
**Confidence:** 4

**Summary:**

This submission is an interpretability case study that seeks to understand how LLMs adjust during multi-modal fusion. Model diffing using SAEs is applied to LLaVA-MORE (with Llama 3.1 8B) to identify which features change after visual instruction tuning. This is achieved by finetuning pretrained LLAMA-SCOPE SAEs to the text tokens of all hidden states. The paper claims to discover "vision" features that emerge after finetuning, specifically those related to spatial positioning and pinpoints them to a few attention heads.

**Strengths:**

- The submission is a novel step to understand the induced feature changes during vision-language adaption.
- The preliminaries cover essential basics to onboard most readers
- The paper is quite didactically written and follows a logical path
- Design choices and experiments are largely well motivated and ablated (except for some crucial parts; see below)
- SOTA interp methods are utilized

**Weaknesses:**

- My biggest concern about this paper is that even if the methodology were flawless (and I am not convinced about that), the scope of the submission is very narrow (effectively, the application of existing methods to trace "spatial" features in Llama 3.1 8B). This might be interesting to a focused interp audience, but is unlikely to address the larger ICLR community. For instance, findings may not generalize beyond the older training styles, which did not fine-tune the encoder–yet, almost all newer models do.
- Causality is not proven: the submission is upfront about not proving causality (e.g., via ablations or steering)–yet this seems to be quite crucial, especially in such a narrow setting.
- The paper is very upfront about the recently discovered failure modes of SAEs, yet, does not seem to provide any countermeasures. For example, if "SAEs [...] are not, however, a complete decomposition: interpretability can vary across runs and training setup" (L143ff), then the study should include multiple SAE training and report CIs.
- The SAEs are only trained on text-tokens. While I understand the empirical reason (pretrained SAEs have a mismatched basis)–this ignores the majority (presumably) of tokens.
- Feature labeling is done via gpt-4o-mini. The details in Appendix B are too coarse to fully understand (e.g., what were the inputs? what was the prompt? generation args?) but either way, a) VLMs are generally quite bad at understanding commonalities/differences between images and b) the model is quite weak. However, it is hard for me to assess if this is a problem without knowing all the details.
- I am not convinced that the shown heads encode the determined relation. For instance, Layer 15, Feature 10748 L13H18 fails to attend to many objects “in front” and rather attends to hands/paws. However, labeling features is a general issue in interp.


Minor:
- Nit: the LLM is by definition NOT a backbone, the vision encoder can be considered one.
- Inconsistent usage of VLM/MLLM: It would be great to settle for one.
- Figures are not vector graphics and are blurry/non-readable upon zoom
- L337: two periods

**Questions:**

The weaknesses listed above mention my concerns.

---

> ### Author Response · Authors · 2025-12-04
>
> Thank you for your thoughtful and detailed review. We appreciate the recognition of our methodological contribution and didactic clarity. Below, we respond to each concern and clarify improvements made in the revised version.
>
> > The scope is narrow (spatial features in Llama-3.1-8B, older training style) and may not generalise to newer LVLMs.
>
> We agree and have revised the positioning accordingly. Our goal is a focused case study of how multimodal fine-tuning reshapes a pretrained linguistic representation, rather than a broad survey across all LVLM families. The methodology itself is architecture-agnostic; it only requires aligned checkpoints and activations, and can directly extend to newer systems as high-quality SAEs become available.
>
> > Causality is not proven; analysis appears correlational.
>
> Thank you for raising this. We have strengthened the emphasis on causal interventions. We want to state that we do ablate specific SAE features and observe substantial drops (≈9–14 points) on spatial reasoning tasks, while generic VQA performance remains largely unchanged. We also use attribution patching to identify the attention heads that causally drive these circuits. We do not claim a fully closed-form mechanism, but the analysis goes beyond correlation and includes explicit causal tests.
>
> > SAE failure modes are acknowledged, but no countermeasures such as multiple runs or confidence intervals.
>
> We fully acknowledge the limitations. Training many full SAE suites for a model of this scale is compute-prohibitive, so instead we focus on robustness of conclusions rather than robustness of individual SAE units. We perform threshold sweeps (Appendix Fig. A.5), show stability in feature-layer distribution, and base insights on aggregate structure across tens of thousands of features.
>
> > SAEs only trained on text tokens, ignoring image tokens.
>
> Visual embeddings produced by the projector are often off-distribution relative to the pretrained LLM’s token space, which leads to unstable rotations and poor monosemanticity when training SAEs over image tokens. This phenomenon has been observed in recent mechanistic analyses of multimodal alignment, e.g., Too Late to Recall: Explaining the Two-Hop Problem in Multimodal Knowledge Retrieval (Venhoff et al., 2025) and Visual Representation Alignment for Multimodal Large Language Models (VIRAL, 2025). We also experimentally attempted SAEs on full sequences (both image and text tokens), but found that the resulting features exhibited poor monosemanticity and higher reconstruction error. In contrast, text-token SAEs preserve consistent feature identity across checkpoints while still allowing cross-modal circuits to reveal visually grounded behaviour using cross-attention.
>
> > Feature labeling with GPT-4o-mini is underspecified and possibly unreliable.
>
> We appreciate this request for clarity and now include the exact prompts, input formats, and generation settings in Appendix B. Importantly, we rely on labeling only for interpretability — not for any quantitative result. All core statistics (frequency shifts, ablations, attributions) are entirely model-based. We also manually verify the spatial features highlighted in the main text.
>
> > Not convinced the shown heads encode the relation (example: L13H18).
>
> Thank you, that is actually intentional. The example cited is a bottom-ranked head and is shown as a negative control to illustrate what weak attribution looks like (diffuse attention, incorrect focus). We will clarify this in the caption and expand comparisons between top- and bottom-ranked heads.
>
> > Minor issues: terminology, blurred figures, punctuation.
>
> We now use consistent terminology, have replaced all figures with vector graphics, and fixed punctuation issues including the double period at L337.

---

### Official Review · Reviewer_faLR · 2025-11-01

**Soundness:** 3
**Presentation:** 3
**Contribution:** 2
**Rating:** 6
**Confidence:** 4

**Summary:**

This paper provides a mechanistic interpretability study on how pretrained language backbones adapt after multimodal fine-tuning. Using stage-wise model diffing with sparse autoencoders (SAEs), the authors analyze how features evolve when visual inputs are introduced into a language model. They identify a subset of features that reorient during multimodal fine-tuning and are selectively activated by spatial-related queries. Through attribution patching, they further trace these features to a small set of mid-layer attention heads responsible for spatial reasoning.

**Strengths:**

1. The paper explores an important question about how large vision language models acquire spatial reasoning capabilities by going beyond standard probing to conduct mechanistic-level analysis.
2. It extends stage-wise model diffing to the multimodal domain, representing a clear methodological contribution.
3. The combination of feature-level diffing, attribution patching, and ablation-based causal validation provides a systematic and interpretable pipeline for studying model internals.

**Weaknesses:**

1. The generality and scalability remain limited. Current mainstream LVLMs (Qwen2.5/3-VL, LLaVA-OV-1.5) usually have different model structures (native resolution image and improved multimodal RoPE) and more complex training stages compared with the selected model in the study. The experiments are conducted only on LLaVA-More with an LLaMA-3.1-8B backbone, leaving it unclear whether the findings generalize to other LVLM architectures or training pipelines.

2. The paper’s focus on spatial reasoning is narrow and may not reflect the broader adaptation patterns involved in multimodal fine-tuning. Many of the results could be dataset-specific, given that the probing relies heavily on VQAv2 and the Visual Spatial Reasoning dataset.

3. The study only compares model behavior before and after multimodal fine-tuning, without examining how representations evolve during the training process. It would be more insightful to analyze how features and attention patterns gradually adapt throughout different training stages and how these changes correlate with the composition or difficulty of the multimodal training data.

**Questions:**

Given that reinforcement learning post-training of multimodal large models has recently become a major focus, could the proposed interpretability framework be extended to compare how features and behaviors evolve under SFT vs. RL fine-tuning? Such an analysis might reveal why RL often leads to stronger generalization than supervised fine-tuning, offering deeper insights into post-training adaptation dynamics.

---

> ### Author Response · Authors · 2025-12-04
>
> We thank the reviewer for this thoughtful summary and these constructive questions. We are glad that the overall approach is recognised as interesting and valuable. Below, we respond to the concerns and clarify our contributions.
>
> > The generality and scalability remain limited. Experiments only on LLaVA-More with LLaMA-3.1-8B backbone; unclear whether findings generalise to other LVLM architectures or training pipelines.
>
> We agree that our evaluation uses a single backbone and setup. We chose LLaVA-More because high-quality SAEs and relevant checkpoints are publicly available, enabling reliable, controlled analysis. The methodology itself is architecture-agnostic: it only requires aligned checkpoints and activations, so in principle it can be applied to other LVLMs as compute allows. We now make explicit that this work should be viewed as a focused case study rather than a sweeping claim of generality.
>
> > The focus on spatial reasoning is narrow — may not reflect broader adaptation behaviors; results might be dataset-specific due to reliance on VQAv2 and a spatial reasoning dataset.
>
> Spatial reasoning was selected as a well-defined, interpretable capability change that allows us to precisely analyse how visual inputs reshape internal representations. However, to support broader applicability, we have now added an OCR case study in the revision: applying the same pipeline recovers a distinct family of visually grounded features related to text-in-image. This suggests that our framework is not tied only to spatial tasks and has the potential to analyse other multimodal behaviours.
>
> > The study compares only pretrained vs. final model; no examination of representation evolution across multiple fine-tuning stages.
>
> Currently, our experiments compare a few regimes (e.g., general-VQA vs. spatial prompts, or pre-finetuned vs. multimodal model), which already provide insight into how features adapt under input distribution shifts. We agree that a dense checkpoint trajectory analysis would be even more informative. This is primarily limited by compute cost (retraining SAEs across many checkpoints). We have added as future work the application of our method across training checkpoints, which would reveal temporal dynamics of feature adaptation.
>
> > Given rising interest in RL post-training, could the interpretability framework be extended to compare SFT vs. RL fine-tuning? That might show why RL leads to stronger generalization than supervised fine-tuning.
>
> Yes — our framework naturally extends to that comparison. One could train SAEs on the base model, then compare across versions fine-tuned via SFT vs. RL. This would reveal which circuits are strengthened or weakened under different regimes, and may explain RL-driven generalisation improvements. We note this as an exciting future direction in the revised discussion.

---

### Official Review · Reviewer_g3sr · 2025-11-03

**Soundness:** 3
**Presentation:** 2
**Contribution:** 3
**Rating:** 4
**Confidence:** 2

**Summary:**

This paper focus on understanding what happens to the language model during multimodal finetuning, it uses SAE as a tool to study stage-wise model diffing. Specifically, authors adapting SAE to LLaVA-MORE's text tokens' hidden states, then identify the adapted features through reorientation and visual energy. Then the authors find the top spatial features by frist comparing the distribution of vqa and spatial, and then further doing lexical artifacts filtering. Based on this method, the authors have some interesting findings: 1) some feature has monosemantic meanings 2) specific head can be located in charing of processing specific information 3) we are able to find features that are crucial for spatial reasoning.

**Strengths:**

1. From the perspective of model diffing, the authors show vivid investigation into how LLMs are adapted to MLLMs
2. The experimental results is convincing, the gap is huge by only turn off single feature.

**Weaknesses:**

1. Figure can be improved, it's not pdf format, can I can hardly tell the details of figure 2.
2. It's not convincing the authors only analyzing text tokens, we cannot train a SAE for image tokens, maybe this is due to the SAE is pre-trained on LLMs, which limits the analysis of vision tokens
3. The author studies the spatial reasoning of MLLM. Is this finding beneficial for improving MLLMs' spatial reasoning capability in benchmarks like What's Up?
4. The application is limited to spatial reasoning, and the feature mining process requires general vqa for contrast, limits the framework to work for general MLLM feature analysis.

**Questions:**

1. What's the process of finding the 146 top spatial features out of 711 candidates?
2. I feels this is highly related to LLM knowledge editing methods, can author discuss the connection with these methods?

---

### Meta-Review · Area_Chair_n7m1 · 2026-01-07

**Summary:**

Reviewers generally liked the idea and tools, and found the mechanistic analysis interesting and competently done. The main hesitation was scope. The paper studies one model, one fine-tuning setup, and mostly one capability, spatial reasoning, which made several reviewers question how general the findings are and whether the contribution is broad enough for ICLR. There were also concerns about SAE robustness, reliance on text tokens only, and early presentation issues. Overall, this pushed the paper into a borderline reject zone. The AC recommends rejection.

**Reviewer Concerns:**

The rebuttal clearly fixed presentation problems, explained the text-only SAE choice with evidence and citations, clarified the feature selection process, and strengthened the causal story using ablations and attribution.

Repositioning the work as a case study and adding an OCR example also helped.

Still, core limitations remain. The work is still based on a single model and training pipeline, SAE stability is argued rather than empirically demonstrated with multiple runs, and broader generalization across architectures and training stages is left as future work.

**Reviewer Scores:**

Reviewer g3sr would likely bump their score up, since most of their concerns were directly addressed and they were already positive.
Reviewer faLR would probably stay about the same, as their main concerns were acknowledged but not fully resolved.
Reviewer 9Shp might move slightly upward due to clearer causality and fixes, but would likely remain skeptical about scope and impact. Reviewer WZEM could also increase a bit due to improved clarity and robustness checks, though core generality concerns would persist.

---

### Decision · Program_Chairs · 2026-01-26

Reject